Replacement of native by non-native animal communities assisted by human introduction and management on Isla Victoria, Nahuel Huapi National Park

Martin-Albarracin Valeria L. 1 valemartinalba@gmail.com
Nuñez Martin A. 2
Amico Guillermo C. 1
1 Laboratorio Ecotono, INIBIOMA, CONICET-Universidad Nacional del Comahue , Bariloche, Río Negro , Argentina
2 Grupo de Ecologia de Invasiones, INIBIOMA, CONICET-Universidad Nacional del Comahue , Bariloche, Río Negro , Argentina
Traveset Anna
Electronic publication date: 2015 Oct 20
Publication date: 2015
Volume: 3
Electronic Location ID: e1328
Received 2015 Jul 14; Accepted 2015 Sep 24
Copyright: © 2015 Martin-Albarracin et al.
Copyright year: 2015
Copyright holder: Martin-Albarracin et al.
License: This is an open access article distributed under the terms of the Creative Commons Attribution License, which permits unrestricted use, distribution, reproduction and adaptation in any medium and for any purpose provided that it is properly attributed. For attribution, the original author(s), title, publication source (PeerJ) and either DOI or URL of the article must be cited.
License URL: https://creativecommons.org/licenses/by/4.0/

Keywords: Invasive species management, Protected areas, Invasive mammals, Introduced deer

Funding: Agencia Nacional de Promoción Científica y Tecnológica FONCyT PICT 2008 2242 National Science Foundation of the USA DEB 948930 Consejo Nacional de Investigaciones Científicas y Técnicas This study was funded by Agencia Nacional de Promoción Científica y Tecnológica (FONCyT PICT 2008 2242) and National Science Foundation of the USA (award DEB 948930). VM was supported by a PhD fellowship from Consejo Nacional de Investigaciones Científicas y Técnicas (CONICET) during this project. The funders had no role in study design, data collection and analysis, decision to publish, or preparation of the manuscript.

==============================
One of the possible consequences of biological invasions is the decrease of native species abundances or their replacement by non-native species. In Andean Patagonia, southern Argentina and Chile, many non-native animals have been introduced and are currently spreading. On Isla Victoria, Nahuel Huapi National Park, many non-native vertebrates were introduced ca. 1937. Records indicate that several native vertebrates were present before these species were introduced. We hypothesize that seven decades after the introduction of non-native species and without appropriate management to maintain native diversity, non-native vertebrates have displaced native species (given the known invasiveness and impacts of some of the introduced species). We conducted direct censuses in linear transects 500 m long (n = 10) in parallel with camera-trapping (1,253 camera-days) surveys in two regions of the island with different levels of disturbance: high (n = 4) and low (n = 6) to study the community of terrestrial mammals and birds and the relative abundances of native and non-native species. Results show that currently non-native species are dominant across all environments; 60.4% of census records and 99.7% of camera trapping records are of non-native animals. We detected no native large mammals; the assemblage of large vertebrates consisted of five non-native mammals and one non-native bird. Native species detected were one small mammal and one small bird. Species with the highest trapping rate were red and fallow deer, wild boar, silver pheasant (all four species are non-native) and chucao (a native bird). These results suggest that native species are being displaced by non-natives and are currently in very low numbers.

Introduction

Although the invasion by non-native species is currently recognized as one of the main threats to global biodiversity, historically this was not always the case. Throughout human history there have been many intentional introductions with the aim of naturalizing species considered valuable. For example, in Hawaii 679 species were purposely introduced and released between 1890 and 1985 for the biological control of pest species; 243 (35.8%) of them have become established (Funasaki et al., 1988). Similarly, in the United States 85% of 235 woody species naturalized were introduced primarily for the landscape trade and 14% for agriculture or production forestry (Reichard, 1994). Interestingly, many populations of non-native species known to have impacts are currently not managed or are protected because they constitute economic resources or have cultural importance (Lambertucci & Speziale, 2011; Nuñez & Simberloff, 2005). Moreover, intentional attempts to introduce new species are still common (Hulme et al., 2008). Together, these factors contribute to the colonization and success of invasive species.

One main objective of protected areas is the protection of native biodiversity (Naughton-Treves, Holland & Brandon, 2005). However, without appropriate management, establishment of a protected area is not enough to protect native biodiversity (Leverington et al., 2010). Biological invasions in particular are an important threat to protected areas because they can have large impacts on native species (Simberloff et al., 2013). Moreover, an invasive species can be unintentionally introduced or reach a protected area by spreading from other sites (e.g., Fasola et al., 2011). If not controlled, these species can increase in abundance and become a serious problem.

We can expect three different scenarios as the result of the introduction of non-native species (MacDougall, Gilbert & Levine, 2009). One is a scenario where non-native species do not survive or are reduced to very low numbers, possibly owing to biotic resistance from native species (Zenni & Nuñez, 2013). Another is where native and non-native species coexist, which could be explained, for example, by the existence of empty niches that are filled by non-native species (Azzurro et al., 2014). The third scenario is where natives are gradually driven to extinction and replaced by non-natives (Blackburn et al., 2004; Woinarski, Burbidge & Harrison, 2015). In this last scenario biological invasions become a very important threat to native biodiversity.

In Andean Patagonia, southern Argentina and Chile, biological invasions are a serious problem, even in national parks (Sanguinetti et al., 2014), where the highest invasion indices have been recorded in relation to other protected areas in Argentina (Merino, Carpinetti & Abba, 2009). For example, studies on the diet of an assemblage of native carnivores found that their diet comprises almost exclusively non-native animals, indicating that these have replaced native species as a food source for native carnivores (Novaro, Funes & Walker, 2000). Similarly, the diet of the condor (Vultur gryphus), a scavenging bird of South America, was historically dominated by guanacos (Lama guanicoe) and lesser rheas (Rhea pennata), the dominant herbivores of the region, but now has shifted and comprises mainly non-native species (Lambertucci et al., 2009). Research in forests and shrublands of Patagonia shows that terrestrial communities are dominated by non-native mammals, including several invasive species such as Cervus elaphus (red deer), Sus scrofa (wild boar) and Lepus europaeus (European hare, Gantchoff, Belant & Masson, 2014). Moreover, the association of some of these species to human-disturbed environments such as roads or pine plantations can increase their rate of spread (Gantchoff, Belant & Masson, 2014; Lantschner, Rusch & Hayes, 2012). Many of these data suggest that the actual problem is not a single species invasion, but a multi-species invasion.

Isla Victoria, located in the centre of Nahuel Huapi National Park, has a history of invasions, with many species of plants and animals actively introduced for several decades (Simberloff, Relva & Nuñez, 2002). In 1937 a zoological station was established on the island with the aim of exhibiting native and exotic fauna to tourists and promoting hunting (Daciuk, 1978a). Non-native species included some of the most invasive vertebrates in the world; such as red deer (C. elaphus), fallow deer (Dama dama), and several pheasant species. The zoological station closed in 1959 and animals were released. Since then, non-native species had not received any significant management, though various proposals have been advanced occasionally (Daciuk, 1978a). Recently, records of native mammals on Isla Victoria have diminished drastically, and those of non-native mammals have become common.

The aim of this study is to assess the community composition of terrestrial mammals and birds in Isla Victoria several decades after the introduction of non-native species. Specifically, we recorded all species detected, and, for the more common species, we estimated the population density and their association with different disturbance levels. We hypothesize that (1) without appropriate management, non-native species have become dominant; and (2) that given the adaptation of some non-native species to human altered habitats, highly disturbed areas will harbor greater abundances and diversity of non-native animals than less disturbed areas.

Methods

Study site

The study was conducted in Isla Victoria, located in the core of Nahuel Huapi National Park, in the northern Patagonian Andes, Argentina (40°57S, 71°33W, APN research permit N°1146). This island is located in the center of Nahuel Huapi Lake, a glacial lake with 557 km2 surface that is located at an altitude of 770 masl. The island has a surface of 31 km2 and a maximum altitude of 1,050 masl. The climate is cold and temperate with a pronounced seasonality. The island is dominated by forests of native Nothofagus dombeyi (Coihue) and Austrocedrus chilensis (Ciprés) (Simberloff, Relva & Nuñez, 2003). Since the beginning of the 20th century, this island has been the focus of many animal and plant introductions, most of them conducted for economic purposes.

Old World deer C. elaphus (red deer), D. dama (fallow deer), and C. axis (axis deer) were introduced to this region between 1917 and 1922 as game animals (Simberloff, Relva & Nuñez, 2003). In 1937 a zoological station was constructed to raise animals for exhibition to tourists and to promote hunting (Daciuk, 1978a). The first two deer species successfully established and are common in the island (Relva, Westerholm & Kitzberger, 2009), while the last one became extinct. Several species of phasianids were also introduced, including peacocks (Pavo cristatus), golden pheasants (Chrysolophus pictus), silver pheasants (Lophura nycthemera), dark pheasants (Phasianus sp.), and ring-necked pheasants (Phasianus colchicus) (Daciuk, 1978a).

Sus scrofa (wild boar) was seen for the first time in the island in 1999. This species was introduced to Patagonia in the early 1900s and probably reached the island swimming from the nearby Huemul Peninsula (Simberloff, Relva & Nuñez, 2003; see Fig. 1), although is also possible that it was illegally and covertly introduced. They are now reproducing regularly and are widespread along the island (Barrios-Garcia, Classen & Simberloff, 2014). Other non-native species more recently established on the island is Neovison vison (American mink), introduced to Patagonia in 1940s and currently spreading (Fasola et al., 2011). Domestic cats, F. domesticus, were brought to the island by the first settlers (date unknown). Several cats escaped from domestication and are now living and reproducing in a wild state. Non-native rodents of the genera Mus and Rattus can be found in the most intensively used ports of the island—Anchorena, Piedras Blancas, and Radal (Fig. 1). Several of these non-native species have been introduced in other regions of the world with reported ecosystem impacts (Barrios-Garcia & Ballari, 2012; Fasola et al., 2011; Relva, Nuñez & Simberloff, 2010; Woinarski, Burbidge & Harrison, 2015). The list of species introduced and naturalized on Isla Victoria is presented on Table 1.

Figure 1 Study area.

Map of Isla Victoria showing the main ports and the transects for camera trapping and censuses. A solid line indicates that both camera trapping and censuses were conducted; a dashed line indicates that only direct censuses were conducted.

Table 1 Introduced and naturalized terrestrial vertebrates on Isla Victoria.

List of introduced and naturalized terrestrial vertebrates on Isla Victoria, Nahuel Huapi National Park; their estimated date of introduction and current status. Pudu pudu was the only native species introduced to the island.

Species	Estimated date of introduction	Current status on the island	
Birds			
Pavo cristatus	1951–1959 (Daciuk, 1978a)	Extinct	
Chrysolophus pictus	1951–1959 (Daciuk, 1978a)	Extinct	
Chrysolophus amhersti	1951–1959 (Daciuk, 1978a)	Extinct	
Lophura nycthemera	1951–1959 (Daciuk, 1978a)	Naturalized	
Phasianus colchicus	1951–1959 (Daciuk, 1978a)	Extinct	
Phasianus sp.	1951–1959 (Daciuk, 1978a)	Extinct	
Numida meleagris	1951–1959 (Daciuk, 1978a)	Extinct	
Mammals			
Rattus sp.	Unknown (Daciuk, 1978a)	Unknown	
Mus sp.	Unknown (Daciuk, 1978a)	Unknown	
Cervus elaphus	1917–1922 (Daciuk, 1978a)	Naturalized	
Cervus axis	1917–1922 (Daciuk, 1978a)	Extinct	
Dama dama	1917–1922 (Daciuk, 1978a)	Naturalized	
Pudu pudu (native)	1951–1959 (Daciuk, 1978a)	Extinct	
Sus scrofa	∼1999, natural spread from continent (Simberloff, Relva & Nuñez, 2003)	Naturalized	
Neovison vison	Unknown, natural spread from continent (Pozzi & Ramilo, 2011)	Naturalized	
Felis domesticus	Unknown (Daciuk, 1978a)	Naturalized (feral)	

The original assemblage of native terrestrial vertebrates on Isla Victoria was relatively simple, consisting on a subset of few species respective to the total fauna of Nahuel Huapi National Park (Table 2) (Grigera, Úbeda & Cali, 1994). It was composed of several lizards of the genus Liolaemus, a snake, two terrestrial birds and some small mammals (Contreras, 1973; Daciuk, 1978b). Two native cervids were observed on Isla Victoria in early 1900s, Pudu pudu (pudú) and Hippocamelus bisulcus (Austral huemul, Daciuk, 1978b; Koutché, 1942). References indicate that H. bisulcus was common on the island at the beginning of the 20th century. Remains of this species have been found in excavations at Puerto Tranquilo, in the north coast of the island (E Ramilo, pers. comm., 2015). On the contrary, there is not enough evidence to say that P. pudu inhabited the island. Instead, individuals observed probably reached the island from populations surrounding Nahuel Huapi Lake (E Ramilo, pers. comm., 2015). Pudu pudu was introduced to the island at the zoological station, and some recent sightings indicate that it is still present, although it appears to be very scarce; however, there are no reported sightings of H. bisulcus from the last decades (http://www.sib.gov.ar/area/APN*NH*Nahuel%20Huapi#eves).

Table 2 Original assemblage of native terrestrial vertebrates on Isla Victoria.

List of species of the original assemblage of native terrestrial vertebrates on Isla Victoria, Nahuel Huapi National Park according to historical records and their current status.

Species	Reference of historical records	Current status on Isla Victoria	
Reptiles			
Liolaemus spp.	Daciuk (1978b)	Present (V Martin-Albarracin, pers. obs., 2015)	
Tachymenis chilensis	Not available	Present (V Martin-Albarracin, pers. obs., 2015)	
Birds			
Scelorchilus rubecula	Daciuk (1978b)	Present (this study)	
Pteroptochos tarnii	Daciuk (1978b)	Probably Extinct	
Mammals			
Dromiciops gliroides	Daciuk (1978b)	Present (D Rivarola, pers. comm., 2015)	
Oryzomys longicaudatus	Contreras (1973)	Present (this study)	
Irenomys tarsalis	Not available	Present (D Rivarola, pers. comm., 2015)	
Hippocamelus bisulcus	Koutché (1942)	Extinct	

Sampling design

To study the composition of the community of terrestrial vertebrates in Isla Victoria we installed one camera trap in each of eight 500 m-transects from winter 2011 to autumn 2012 (Fig. 1). Transects were associated with two different levels of human disturbance: high (4 transects) and low (4 transects). High disturbance occurred in regions where tourist activities are developed, with tens to hundreds of people walking along paths daily. These regions include plantations of non-native trees and shrublands with high abundances of non-native plants. Regions with low disturbance were occasionally visited by people who inhabit the island. These regions are dominated by forests of N. dombeyi and A. chilensis and by mixed shrublands dominated by native plants. We used eight heat and motion-triggered infrared cameras; six were model Bushnell Trophy Cam 119736C (Bushnell, Overland Park, Kansas), and the other 2 were Stealth Cam Unit IR (Stealth Cam, Grand Prairie, Texas). Cameras were located haphazardly along transects, installed at a height of 30–50 cm and programmed to take videos 40 s long with a 1-min delay between exposures. Locations of the cameras were chosen based on visibility, but we did not seek animal trails (Rowcliffe et al., 2008). After 2–5 weeks, videos were downloaded and cameras were relocated along transects at new sites. The overall effort was of 1,253 camera days (minimum camera days per transect = 86; maximum = 289; average = 156.6).

In addition to camera-trapping, direct census of animals was conducted through a distance-sampling approach. The sampling was conducted using the same eight transects as with the camera traps, plus two extra transects (N = 10) located in low-disturbance areas (Fig. 1). We walked the transects 3–5 times at an average speed of 2 km per hour recording all the terrestrial mammals and birds detected (sighted or heard), and their perpendicular distances to the transect. For further analyses, perpendicular distances were truncated at 0, 5, 10, 15, 20, 30, 50, 100 and 150 m.

Data analysis

Study of habitat use

To compare habitat use at high vs. low disturbance sites, for each of the most frequently captured animals (deer, pheasant, and boar) we calculated the relative abundance index (RAI). This index was calculated as the number of independent captures obtained through camera-trapping (C) divided by trapping effort (TE) and multiplied by 100 camera-days. We considered captures of the same species as independent only when there was a difference of at least one hour between captures of studied species. RAI=CTE*100camera-days.

As the effective trapping area differs widely among species with different body size (Rowcliffe et al., 2011), RAI was not used to make comparisons between species.

Estimation of population density

Population density was estimated from distance sampling data (Buckland et al., 2007). The half-normal (HN), hazard-rate (HAZ) and exponential (EXP) key functions for detection probability were fitted to truncated data of distance. For species with over 30 sightings (L. nycthemera and S. rubecula), we used type of environment (plantation, forest and shrubland) as a covariate of detection probability and level of disturbance (high or low) as a covariate of density (Marques et al., 2007). For the other species we used no covariates. The Akaike Information Criterion (AIC) together with diagnostic plots were used to choose between models (Appendix S1).

For each species we conducted a regression of log observed cluster size vs. estimated detection probability to test for size bias (i.e., tendency to observe larger clusters at longer distances). In all cases, the regression slope was not significantly different from zero (P > 0.46). We thus used mean observed cluster size as an estimate of expected cluster size to calculate animal densities. Cluster size data were obtained from camera trap videos.

Results

Habitat use

We obtained a total of 710 independent captures of 8 mammal and terrestrial bird species through camera trapping. The species detected included one native small mammal (O. longicaudatus), five non-native mammals (C. elaphus, D. dama, S. scrofa, F. domesticus and N. vison), one native bird (S. rubecula) and one non-native bird (L. nycthemera). The great majority of captures (99.7%) were of non-native animals (Fig. 2). The species detected most frequently were non-native deer (55.4% of captures including both species), L. nycthemera (31.2%), and S. scrofa (11.3%). We did not capture other native species reported in the island by Daciuk (1978b). Deer captures were identified at species level when possible (83% of captures; 65% corresponded to C. elaphus and 35% to D. dama).

Figure 2 Camera trapping captures of terrestrial species.

Total number of captures obtained by camera trapping for each terrestrial species, including species reported for the island in Daciuk (1978b) and not detected in this study.

At highly disturbed sites we detected seven species, while at low disturbance sites we observed five. The only species detected in low disturbance sites but not in highly disturbed regions was N. vison. Three species were detected only in highly disturbed regions: F. domesticus, S. rubecula and O. longicaudatus (but the last two had only one capture each). Lophura nycthemera was relatively more abundant in highly disturbed areas than in low disturbance areas (Pearson’s chi-squared test, p = 0.001; Fig. 3), while relative abundance indices of deer and S. scrofa did not vary among sites.

Figure 3 Habitat use.

Relative abundance index (RAI) for the four species most frequently captured in areas with high and low levels of disturbance. N represents the total number of captures obtained for each species. An asterisk indicates species with differential use of low and high-disturbed habitats.

Figure 4 Detectability of animals.

Histograms of observed distances and fitted detection functions for L. nycthemera (A and B respectively, N = 33), S. rubecula (C and D, N = 36), C. elaphus and D. dama (E and F, N = 15), and S. scrofa (G and H, N = 7).

Population density

We detected five terrestrial animals through direct censuses: L. nycthemera, C. elaphus, D. dama, S. scrofa and S. rubecula. For L. nycthemera, the best model fit was achieved by the exponential key function, with type of environment as a covariate for detection and level of disturbance as a covariate for density. Lophura nycthemera density was nearly twice as high (1.79 ± 0.52 ind/ha) in highly disturbed areas than in low disturbance areas (0.99 ± 0.42 ind/ha). For S. rubecula, best fit model used the hazard rate key function and level of disturbance as a covariate for density. S. rubecula density was more than twice as high in low disturbance areas (0.73 ± 0.19 ind/ha) than in highly disturbed areas (0.32 ± 0.12 ind/ha). For deer, the best fit model used the hazard rate key function, and estimated density was 0.12 ± 0.05 ind/hectare. For S. scrofa, the best fit model used the exponential key function and estimated density at 0.27 ± 0.16 ind/hectare (Fig. 4).

Discussion

Our results show that at least six non-native species have successfully established on Isla Victoria and become dominant. By contrast, we found very few native species. Non-native deer (C. elaphus and D. dama), S. scrofa, and L. nycthemera were among the most abundant species on Isla Victoria, and only L. nycthemera showed greater abundance in highly disturbed areas than in low disturbance areas. While it is difficult to assess whether non-native species are displacing native species (because there is no quantitative information of native populations before non-native species introductions), it is likely that the successful establishment of the non-natives could have contributed to their decline.

The current assemblage of non-native animals on the island results from a combination of intentional and unintentional introductions of species and range expansions of invasive species from the continent. Old World deer and pheasants, for example, were intentionally introduced. Non-native rodents present on the island may have been introduced unintentionally through transport in the hold of ships. Some species that are believed to have reached the island by expansion of their invading ranges are S. scrofa and Neovison vison.

Several factors can influence the likelihood that an introduced species will become established. On Isla Victoria many factors such as species traits, propagule pressure, and climatic matching have helped non-native species invasions. Cervus elaphus, D. dama, S. scrofa, P. colchicus, and N. vison are known to have specific traits that make them good invaders in many regions (Table 3). In addition, the introduction of species bred in the zoological station was not a unique event, but animals were released continuously for several years, increasing propagule pressure and therefore increasing the likelihood of successful establishment (Lockwood, Cassey & Blackburn, 2005). Moreover, the area possibly offered an empty niche for some species (Azzurro et al., 2014). The absence of big predators, for example, may have aided naturalization by non-native vertebrates. Similarly, land birds in Patagonian forests are small in relation to non-native pheasants, which may have different requirements. All these factors made Isla Victoria an ideal site for species naturalization. As result of these multiple invasions, together with inadequate management, the current assemblage of terrestrial mammals and birds on the island is highly dominated by non-native species, in both composition and abundance.

Table 3 Antecedents of invasion of non-native species detected on Isla Victoria.

List of non-native species detected on Isla Victoria and their known native range, invaded regions and impacts reported.

Species	Native range	Invaded regions	Known impacts	Reference(s)	
Cervus elaphus	Eurasia	North and South America, New Zealand and Australia.	Impact on natural regeneration of the native forest and facilitation of non-native plant growth. Dispersal of non-native ectomycorrhizal fungi that promote Pinaceae invasions. Competitive displacement of native deer.	Barrios-Garcia, Relva & Kitzberger (2012), Coomes et al. (2003), Nuñez et al. (2013), Nuñez, Relva & Simberloff (2008), Relva, Nuñez & Simberloff (2010) and Wood et al. (2015)	
Dama dama	Eurasia	North and South America, South Africa, New Zealand and Australia.	Impact on natural regeneration of the native forest and facilitation of non-native plant growth. Dispersal of non-native ectomycorrhizal fungi that promote Pinaceae invasions. Competitive displacement of native deer.	Barrios-Garcia, Relva & Kitzberger (2012), Nuñez et al. (2013), Nuñez, Relva & Simberloff (2008) and Relva, Nuñez & Simberloff (2010)	
Sus scrofa	Eurasia, north of Africa	Widely distributed worldwide, it is present on all continents except Antarctica, and many oceanic islands.	Change in soil structure and processes, reduction of plant cover, decreasing of plant species diversity, alteration of plant species composition, predation of seeds of native species, increase of non-native plants abundance. Predation, nest and habitat destruction, and resource competition with other animals. Dispersal of non-native ectomycorrhizal fungi that promote Pinaceae invasions. Alteration of water quality and chemistry.	Barrios-Garcia & Ballari (2012), Barrios-Garcia, Classen & Simberloff (2014), Barrios-Garcia & Simberloff (2013), Massei & Genov (2004) and Nuñez et al. (2013)	
Lophura nycthemera	Southeast Asia	Argentina and Germany.	Competition with native fauna, seed dispersal of non-native plants.	Daciuk (1978a) and Lever (2005)	
Felis domesticus	Domesticated from the Wildcat (F.s. lybica), probably 9–10,000 years ago in the Fertile Crescent region of the Near East.	Widely distributed worldwide, it is present on all continents except Antarctica, and many oceanic islands	Predation on native fauna including reptiles, birds and mammals. Responsible for many extinctions on oceanic islands.	Driscoll et al. (2007), Loss, Will & Marra (2013), Medina et al. (2011) and Woinarski, Burbidge & Harrison (2015)	
Neovison vison	North America	Argentina, Chile, widely distributed throughout Eurasia.	Predation on native fauna including mammals, birds, amphibia and Crustacea. Competition with native minks.	Bonesi & Palazon (2007) and MaCdonald & Harrington (2003)	

Disturbance has long been cited as a factor that helps non-native species colonization and invasion (Hobbs & Huenneke, 1992). Thus, the association of non-native animals with highly-disturbed areas such as conifer plantations could be facilitating the invasion of natural areas by non-native herbivores (Lantschner, Rusch & Hayes, 2012). Our results showed that the only non-native animal that consistently associated with highly disturbed areas was L. nycthemera. Cervus elaphus, D. dama and S. scrofa, instead, made a similar use of low and high disturbance areas. Previous studies in the area showed that C. elaphus and S. scrofa prefer pine plantations instead of native vegetation at the landscape scale (Lantschner, Rusch & Hayes, 2012) and revealed a positive association of S. scrofa with roads (Gantchoff & Belant, 2015). These results and our study thus suggest that deer, S. scrofa, and L. nycthemera are highly capable of using human-disturbed habitats. While deer and S. scrofa can also reach high abundances in native environments, and for instance have large impacts on native species inhabiting them, L. nycthemera may remain strongly associated with human-disturbed environments and scarce in native environments. S. rubecula, the only native land bird detected, was strongly associated with low disturbed environments. This could simply be due to the preference for native habitats (Lantschner & Rusch, 2007), but it is also possible that the pheasant is displacing it from plantations. Pteroptochos tarnii coexists with S. rubecula in all areas surrounding Isla Victoria (Amico, García & Rodríguez-Cabal, 2008), but it was not detected in this study. We hypothesize that both human disturbance and the presence of non-native species may be affecting P. tarnii abundance (Lantschner & Rusch, 2007; Skewes, Rodriguez & Jaksic, 2007).

One fact that can have a big impact on native biodiversity is the naturalization of non-native terrestrial carnivores, N. vison and F. domesticus, because the original assemblage of vertebrates on isla Victoria had no terrestrial carnivores (see Table 2). Thus, these species can have an important role as predators of birds and small mammals. A species that can be seriously affected by the naturalization of N. vison is imperial shag, Phalacrocorax atriceps, a species that nests at steep rocky cliffs of the island and that is considered of special value by the National Park Service (Pozzi & Ramilo, 2011). Neovison vison and F. domesticus can also be involved in the apparent local extinction of P. tarnii and can threat populations of other native ground-nesting birds such as S. rubecula.

We must take into account that the low number of native species detections may be partially explained by the low body mass of some species (Dromiciops gliroides, Oligoryzomys longicaudatus, S. rubecula and P. tarnii). Trap cameras have a bigger effective trapping area for species of higher body mass; for example in Barro Colorado Island studies found that effective detection distance is about 1.3 m for species of low boy mass (mouse unknown species, body mass = 0.1 kg) and about 3.5 m for species of higher body mass (Collared peccary Tayassu tajacu, body mass = 25.2 kg) (Rowcliffe et al., 2011). However, camera-trapping has been used successfully for the study of small birds and mammals (Kays et al., 2011). In our study, through direct census S. rubecula was much more frequently detected—usually by its song—than big mammals, and estimated densities exceeded those of deer and S. scrofa. P. tarnii, however, also has an identifiable song but was never detected in direct censuses. Some evidence derived from mouse-trapping campaigns also suggests that small mammals are scarcer on Isla Victoria than on nearby continental areas of the National Park (Nuñez et al., 2013). We also understand that we are considering a snapshot of the abundance of terrestrial birds and mammals, although we believe this pattern is likely to be consistent in time.

It has been suggested that invasive species with no negative impacts on native biodiversity, can be beneficial because they can increase local biodiversity (Thomas & Palmer, 2015) or supply benefits such as habitat or food to native species (Davis et al., 2011). Some invasive species can also have an important role as dispersers of seeds of native species in their introduced range (Chimera & Drake, 2010). On Isla Victoria C. elaphus, S. scrofa and L. nycthemera are consumers of fleshy fruits and might be contributing to seed dispersal of native plants. However, in our study site non-native species have reached such high proportions (see Fig. 2) that we can hypothesize they are displacing native fauna. The replacement of native fauna by non-native animals can have other important consequences for the functioning of local ecosystems. For example, on Isla Victoria, it has been demonstrated that non-native deer prefer native plants rather than non-natives, a fact that could promote the invasion by non-native conifers (Nuñez, Relva & Simberloff, 2008). The consumption of fruits of invasive shrubs by non-native animals, for example fruits of Juniperus communis, Rosa rubiginosa or Rubus ulmifolius, can be promoting plant invasions. Furthermore, soil disturbance by S. scrofa can facilitate invasive plant establishment (Barrios-Garcia & Simberloff, 2013). Lastly, both S. scrofa and deer are involved in the dispersal of mycorrhizal fungi that allow colonization by non-native conifers (Nuñez et al., 2013).

It is difficult to know whether the presence of non-native animals was the driver of native species decline, but based on previous evidence it is likely that at least the successful establishment of the non-natives could have contributed to the decline of the natives. We believe this is likely based on the extremely low capture rate of native vertebrates (0.3% of camera-trapping captures) and the presence of species known to have reduced populations or extinguished species elsewhere (Table 3).

Recently, Nahuel Huapi National Park has started implementing a management plan for invasive non-native vertebrates (Disposition 422/2014, Mujica, 2014). Specifically, this plan regulates the control through hunting of C. elaphus, D. dama, and S. scrofa on Isla Victoria, Nahuel Huapi National Park. This program allowed the removal of more than 150 individuals during its first year of implementation (unpublished data), and it could represent a first step towards the recovering of native biodiversity on Isla Victoria. We suggest that monitoring through camera trapping using a sampling design similar to ours could be an economic way to evaluate the results of this program. Also, we strongly recommend that a plan for the control and eradication of non-native species on the island should also consider N. vison and F. domesticus, because of their role as predators of native fauna. Together with the monitoring of terrestrial fauna using camera-trapping, we suggest conducting a monitoring focused on small mammals (rodents and marsupials) using some array of live traps. In addition, active efforts to reintroduce native deer species as P. pudu and H. bisulcus could be highly beneficial for their global conservation, given that Isla Victoria has proved to be an ideal place for the acclimatization of herbivores, and that these two species are categorized as vulnerable and endangered respectively by the IUCN. Administrators of protected areas should also take measures to prevent the expansion of invasive species and the introduction of new ones to other regions of Nahuel Huapi National Park.

Supplemental Information

Appendix S1 Model selection

Models fitted for Pheasant, Chucao, Deer and Boar and their corresponding AIC values. Functions: half-normal (hn), hazard-rate (haz), exponential (exp); detection covariates: type of environment (Env); density covariates: level of disturbance (Dis).

Click here for additional data file.

Supplemental Information 2 Raw data on camera trapping captures for terrestrial species

Dataset showing data on camera trapping captures for terrestrial species on Isla Victoria. Each row is an independent capture of an animal. Columns describe the species detected, the transect sampled, the disturbance level, the number of individuals (group size) and the type of environment.

Click here for additional data file.

Supplemental Information 3 Raw data on distance sampling for vertebrate species

Dataset showing data on distance sampling for vertebrate species on Isla Victoria. Each row is an animal detection. Columns describe the transect sampled, the species detected, the type of environment, the transect length, the number of individuals (group size) and the perpendicular distance of the animal to the linear transect.

Click here for additional data file.

We are very grateful to the National Park Service of Argentina for help with logistics and research permits and to Cau–Cau for help with transportation to the island. N González, F Villalba, M Mansilla and several national park volunteers helped with field work. We also thank C Chehébar, E Ramilo, N Barrios-Garcia, D Simberloff, D Hansen, A Traveset and one anonymous reviewer for useful comments and suggestions on previous drafts of the manuscript.

Additional Information and Declarations

Competing Interests

Author Contributions

Field Study Permissions

Data Availability

The authors declare there are no competing interests.

Valeria L. Martin-Albarracin conceived and designed the experiments, performed the experiments, analyzed the data, wrote the paper, prepared figures and/or tables, reviewed drafts of the paper.

Martin A. Nuñez and Guillermo C. Amico conceived and designed the experiments, contributed reagents/materials/analysis tools, wrote the paper, reviewed drafts of the paper.

The following information was supplied relating to field study approvals (i.e., approving body and any reference numbers):

Administración de Parques Nacionales de Argentina provided field research permits (Proyecto No 1146)

The following information was supplied regarding data availability:

FigShare

10.6084/m9.figshare.1533298

10.6084/m9.figshare.1533297.

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
