# Peer review of "Replacement of native by non-native animal communities assisted by human introduction and management on Isla Victoria, Nahuel Huapi National Park"

_PeerJ, doi:10.7717/peerj.1328_

## Round 0.1 · original submission · Major Revisions

Your ms has been reviewed by two referees who are rather positive about it (one more than the other), although both also have important concerns regarding methodology, basic information on the system, and justification of some of the statements in the ms.

I agree with Ref 1 that you need to include standard information on Isla Victoria (age, size, latitude, etc.). You need also to include more references that strengthen the statements on historical records, even if they are unpublished reports. This referee makes a number of corrections of typos and suggests some style changes that I think you should take into account when preparing the next version. I myself have detected some typographic errors (e.g. ‘assamblage’ rather than ‘assemblage’, etc. so please carefully check the grammar). I think you also need to be consistent with terminology, so for instance use either ‘non-native species’ or ‘introduced species’.

Ref 2 is more critical about your study design so I urge you to justify in your next version of the ms how you have addressed the possibility of spatial auto-correlation. Likewise, you would need to discuss the possible effect of the different vegetation on animal detectability between the highly disturbed and the lowly disturbed sites. I also think you should mention in the discussion that you are considering a snapshot of the abundance of birds and mammals, even though the pattern you find is likely to be consistent in time. Although you clearly point out the negative impacts of the introduced animals on the system, I think you might also consider briefly discussing the possible effects at least on some plant species by dispersing the seeds of native plants (both deer and wild boar are known to disperse seeds); a number of studies have shown that alien dispersers play an important role in the regeneration of native plant communities (e.g. in Hawaii).

Please also consider the suggestion of this referee (and also by ref 1) of including a table that lists all native species previously having been detected on the island as well as the alien species and approximate date of introduction to the island. In methods, please also give more details on the sensitivity of the two types of cameras you use.
Please in the next version clearly indicate where the data will be made available.

I hope the comments are useful to you and I look forward to receiving a revised version of the paper in due course.

Yours sincerely,
Anna Traveset

·

Basic reporting

The authors provide a short and to-the-point study (well done!), which gives valuable information to island researchers/managers – not only to their Southern American island system but of value to many similar systems affected by non-native vertebrates.
I find this study to be a prime example of why I have recently become very excited about PeerJ. What may otherwise have ended up as a short note in a non-open-access journal will now become widely available to researchers & practitioners.

Study site – I’d like to have more standard information included; e.g., Where exactly are we? How large is the island? Maximum altitude? Climate? How large is the lake/fjord (?) it is in, how did the island originate? (natural/man-made after damming of a river?), ...etc. Also a good idea to include some kind of scale bar on Figure 1.

I think it is essential to strengthen the part on historical records, if at all possible. At the moment, the entire section on this (P7, L10-22) starts with “According to various records”, and contains one reference halfway down, with nothing else referencing the other statements. Even if these various records are personal communications or unpublished/grey literature reports cited in the one reference, I’d strongly urge you to include more information here. A clear desription and referencing of the historical documentation for the baseline community of native species is an essential part of your study. Currently, this part is not quite strong and clear enough. One possible solution is to give an overview (table form) of all native spp., including a column with reference(s) related to it (similar to what you have for the non-native ones).

Introduced cats: I didn’t find the given information & coverage of this species to be sufficiently detailed at present, despite it being the only non-native predator in your study. If at all possible, I‘d like to see a brief discussion of this species & its likely impacts on the island in light of your findings. Also a bit confusing that two different names for this species is used in the study (Felis domesticus in Table 1, F. silvestris catus in Figure 2).

Discussion – I completely agree with the overall conclusion that the diverse menagerie of introduced species have likley directly and significantly (=negatively!) affected the native species. I also agree that most speculations about the mechanisms & pathways will be just that – speculations. But I think you could easily strengthen the perspectives of your study a bit. For example, by including specific recommendations for monitoring and experimental studies that could/should be initiated as part of the ongoing and upcoming control and eradication management program. Solid mechanistic data from such studies could at least provide some quantitative evidence for direct and indirect impacts of the non-native species on the native ones.

Experimental design

I found the study to be convincing in its methods and analyses, and have nothing to add here.

Validity of the findings

In relation to PeerJ requirements, I could not find any information about whether & where the data have been made available, and suggest this should be dealt with appropriately before publication.

Additional comments

General comments: All page numbers refer to the pagination provided on the original manuscript pages.

You have a few sentences that start with an abbreviated Latin binomial (e.g., P11,L12). I believe it’s the normal convention to write out names in full when used as the very beginning of a sentence (even sp. already mentioned in text previously). Please check carefully & correct throughout.

Figure 2 – it may be a pdf-artefact, but on my screen it looks as if the last/right-most four native species have negative values/a bar below zero. Please check & remove if this is the case.

Figure 3 – please use Latin names here, too (you do in the other figures/text). Also please revise the legend accordingly; it is confusing to see ‘deer’ and ‘animal’ used to denote both of the introd. spp.

P1,L6 – ‘in Patagonia’ – for geographically challenged readers, it may be nice to include the country/countries, too. Also in the first mention in the main text (P4,L12).
P3,L3 – correct to “invasion by”
P3,L14 – please specify whether you’re only talking about intentional attempts (I assume this to be the case, given the framing of your study, but a good idea to clarify this).
P4,L4 – suggest changing to “reduced to very low numbers” – ‘attain’ is usually used to describe the outcome of a positive/growing trend, or reaching a target. Also, the second part of the sentence only applies to scenarios with survivors; it doesn’t make sense to say that non-survival is the result of biotic resistance from native species? Suggest rewriting – or actually deleting “do not survive”, since this is your third scenario, anyway?
P4,L7-8 – see comment above; you currently have two scenarios that include extinction as an outcome.
P4, L13 – change to “have been recorded”
P4,25 – consider using ‘increase’ rather than ‘augment’?
P5,L6-7 – change to “on the island”
P5, L13-14 – the link is quite useless as a reference, as it does not take the reader directly to a reference for the statement given. Suggest deleting (or replacing with a reference to a published report/paper with this information).
P5, L16-17 – non-standard reference, with not enough information given for who wrote ths ‘Disposition’ and where/who published it? Suggest giving a more standard reference with name of author(s) and an extended reference with title, etc, of this document in the reference list.
P5,L20 – I assume you mean after the introduction of non-native spp? The native ones have presumably been on the island for longer than “several decades”? Please clarify.
P5,L20-21 – Unclear at present; it sounds a bit strange that you only estimate abundances for non-natives, and then proportion (?of what –individuals? Species?) of native & non-native species. Please rewrite to clarify exactly what was recorded in your study.
P7,L17 – Change from “...island, instead...” to “...island. Instead...”
P8,L17 – please include information on min/max & average camera hours per transect.
P9,L11 – change to “...captures of studied species.”
P10,L13-14 – change to “terrestrial bird species”
P11,L13-19 – it looks a bit weird to shorten to ‘ind’, but write out ‘hectare’ in full. Suggest being consistent & saying ‘ind/ha’?
P11,L23 – consistency: you use the term ‘non-native’ about these species until now; why do you suddenly refer to them as ‘invasive’ here? Or do you mean that they are known to be invasive elsewhere? Please clarify & revise accordingly.
P14,L15 – correct ‘that’ to ‘than’.
P15, L6-9 – suggest rephrasing to “It is difficult to know whether the presence of non-native animals was the driver of native species decline, ...”

Reviewer 2 ·

Basic reporting

The English is generally clear and straightforward, but, like all articles, there are some small errors and this one could use some careful technical editing to clean them up.

Overall, the structure is straightforward, and all sections provide relevant background (with a few minor exceptions noted elsewhere in the review).

A fuller table (like Table 1, but more comprehensive), or series of tables listing all relevant bird and mammal (and reptile?) species would be very useful. In the table, species could be listed in several categories, including: aliens originally introduced, aliens still present, natives originally present, natives still present, or something like that. This will be very helpful to readers. Dates of introduction could be provided here.

I cannot determine whether raw data have been made available.

Experimental design

The research question is meaningful: do invasive alien species displace native species? It is not clear that the methods are adequate to address this question.

This may be a rather pedantic point, but, technically, the highly-disturbed areas are all spatially clustered near Anchorena Port, and therefore not independent of each other, so it may not be valid to compare those points to the low-disturbance points statistically. In addition, if the vegetation is different in high-disturbance and low-disturbance sites, how does this affect things like detectability.

Validity of the findings

This study is essentially a snapshot of the abundance of some bird and mammal species on an island in a national park in Argentina. The authors suggest that the presence of numerous alien animals could be responsible for the low abundance of native birds and mammals. However, without any evidence of mechanisms, and with abundance estimates from just one sampling period, this is very difficult to determine. Without being able to test this, the paper is essentially not much more than an estimate of the current abundance of some of the birds and mammals on the island in question.

Additional comments

Format below = page.lines

4.18-19, 6.19-20, and throughout: Provide scientific names (species or, where appropriate, just genera) for plants and animals the first time they are mentioned. This is inconsistent. A table may be a good way to address this.

5.21: If you are estimating the abundance of non-native animals and the proportions of native and non-native animals, then you must also be estimating the abundance of native animals.

711-12: Scelorchilus rubecula (chucao) and Pteroptochos tarnii (huet-huet) are flightless? Surely not.

8.9-17: Please provide more detail regarding the sensitivity of the cameras. At what distance can they detect a rodent vs. a deer? No camera traps “capture” 100% of the animals that pass them, but detection rates vary a lot based on camera settings, animal sizes and distances, etc. All camera traps, In other words, are somewhat biased. To what extent are yours biased more toward capturing some species (e.g., large species) than others? Would S. rubecula or arboreal mammals be reliably detected? I see this has been partially addressed in 14.5-15, where it is noted that camera traps have successfully been used for both small and large animals in other studies, which is true. They key question, though, is whether they can be used to detect and compare large and small animals simultaneously.

8.19-25. Were the distance-sampling methods based entirely on sighting animals, or was sound used as well?

9.10: independence of camera trap captures. It is stated that camera captures of the same species were considered independent only when there was a difference of at least one hour between captures for a given species. Are any of these animal species social (i.e., do they travel in small groups)? If so, several different individuals could trigger the traps in quick succession if a group passed a camera, but they might be counted as a smaller number of individuals if not all were present in the field of view at the same time. How does this affect the analysis?

9.17: Cite a reference for the methods used for estimation of population density. I do not have the background to evaluate this specific analysis.

15.11: In my opinion, the present study does not highlight the problem of using protected areas as a reservoir of biodiversity—at least not if the “protected area” is a place where lots of non-native animals have been deliberately introduced. Most “protected areas” are not like this.

Fig. 1. Add a scale bar.

Fig. 3. Provide species names.

---

## Round 0.2 · accepted · Accept

Dear Dr. Martin-Albarracín,

I thank you for sending your revised ms to PeerJ and I am glad you found the referees' comments and mine useful. I do think your ms has improved substantially and I thus now deem it as acceptable in PeerJ.

Congratulations of this nice piece of work!

Best wishes,
Anna Traveset